



# Shipborne measurements of methane and carbon dioxide in the Middle East and Mediterranean areas and contribution from oil and gas emissions

Jean-Daniel Paris[1,3], Aurélie Riandet[1,*], Efstratios Bourtsoukidis[2,3], Marc Delmotte[1], Antoine Berchet[1], Jonathan Williams[2,3], Lisa Ernle[2], Ivan Tadic[2], Hartwig Harder[2], Jos Lelieveld[2,3]

[1]Laboratoire des Sciences du Climat et de l'Environnement, CEA-CNRS-UVSQ, UMR8212, IPSL, Gif-sur-Yvette, France
[2]Department of Atmospheric Chemistry, Max Planck Institute for Chemistry, Mainz 55128, Germany
[3]Energy, Environment and Water Research Center, The Cyprus Institute, Nicosia, Cyprus

*Now at Aix Marseille Université, CNRS, Avignon Université, IRD, IMBE, Aix-en-Provence, France

*Correspondence to*: Jean-Daniel Paris (jean-daniel.paris@lsce.ipsl.fr)

**Abstract.** The increase of atmospheric methane ($CH_4$) and carbon dioxide ($CO_2$), two main anthropogenic greenhouse gases, is largely driven by fossil sources. Sources and sinks remain insufficiently characterised in the Mediterranean and Middle East areas, where very few in situ measurements area available. We investigated the atmospheric distribution of $CH_4$ and $CO_2$ in the region through shipborne measurement in July and August 2017. High mixing ratios were observed over the Suez Canal, Red Sea and Arabian Gulf, while generally lower mixing ratios were observed over the Gulfs of Aden and Oman. We probe the origin of $CO_2$ and $CH_4$ excess mixing ratio by using correlations with light alkanes and through the use of a Lagrangian model coupled to two different emission inventories of anthropogenic sources. We find that the $CO_2$ and especially the $CH_4$ enhancements are mainly linked to nearby oil and gas (O&G) activities over the Arabian Gulf, and a mixture of other sources over the Red Sea. The isomeric ratio of pentane is shown to be a useful indicator of the O&G component of atmospheric $CH_4$ at the regional level. Upstream emissions linked to oil in the Northern Arabian Gulf seem to be underestimated while gas-related emissions in the Southern Gulf are overestimated in our simulations. Our results highlight the need for improvement of inventories in the area to better characterize the changes in magnitude and the complex distribution of the O&G sources in the Middle East.

## 1. Introduction

Methane ($CH_4$) and carbon dioxide ($CO_2$) are potent anthropogenic greenhouse gases (GHG). The $CH_4$ atmospheric mole fraction has increased by 150% since the pre-industrial era (Saunois et al., 2020). Roughly half of $CH_4$ sources are of natural origin (mainly from wetlands, with contributions from fires and geologic sources). The remainder is anthropogenic, mainly linked to fossil fuels, agriculture (including enteric fermentation in ruminants, manure management and rice paddies) and waste management. The increasing anthropogenic emissions are driven equally by fossil fuel sources and agricultural sources (Jackson et al., 2020).



$CO_2$ concentration in the atmosphere has increased by 47% since the preindustrial era and reached 407.38±0.10 ppm in 2018 (Dlugokencky and Tans, 2019; Le Quéré et al., 2018; Friedlingstein et al., 2019). Its increase is caused primarily by the use of fossil fuel, cement production and land use change, and is partly mitigated by uptake in terrestrial ecosystems and the ocean. Over the last decade (2009-2018) $CO_2$ sources have been dominated by fossil fuels (9.5±0.5 GtC yr$^{-1}$) with a significant source

from land use change (1.5±0.7 GtC yr$^{-1}$). $CO_2$ is removed from the atmosphere by sinks in the ocean (2.5±0.6 GtC yr$^{-1}$) and land ecosystems (3.2±0.6 GtC yr$^{-1}$; Friedlingstein et al., 2020). Atmospheric $CO_2$ is expected to continue its growth despite widespread adoption of climate policies as oil and gas consumption trends suggest a continued increase in fossil $CO_2$ emission (Jackson et al., 2019).

The 2015 Paris agreement has set the objective to limit global temperatures below 1.5°C. To reach this objective, a corridor of

compliant emission pathways has been designed, that require not only strong reductions in $CO_2$ emissions but also in anthropogenic $CH_4$ emissions (Jones et al., 2018; Nisbet et al., 2020). Regarding $CH_4$, a wide array of measures is necessary (Nisbet et al., 2020), with potentially high mitigation impact in the energy (GIE-MARCOGAZ, 2019), agricultural and waste management sectors (Rogelj et al., 2018).

Fossil fuel production and use is responsible for the release of 112 Mt $CH_4$ yr$^{-1}$ to the atmosphere, representing 33% of the

total anthropogenic emission of $CH_4$ (Saunois et al., 2020). Considering fossil fuel emissions alone, 68% of the emissions is linked to oil and gas (O&G) while the rest is associated to coal mining. Emission of $CH_4$ arises at each step from the production site to the consumption site but a large fraction of the net emission is associated with the production, transport and processing (Alvarez et al., 2016). Emission of $CH_4$ occurs either as fugitive emission (leaks from valves, connectors and compressors, intentional venting) or as incomplete combustion during flaring (GIE-MARCOGAZ, 2019).

Large uncertainties remain associated to the magnitude and spatial and temporal distribution of $CH_4$ sources (Saunois et al., 2020). After a pause between 2000 and 2007, $CH_4$ in the atmosphere has resumed its increase. Constraints from current available observations on the respective contribution to its sources and sinks do not allow a definitive explanation of this pattern (Saunois et al., 2017; Saunois et al., 2020; Turner et al., 2019, Nisbet et al., 2020). Schwietzke et al. (2016) highlighted the strong underestimation of O&G emissions in current inventories (see also Saunois et al., 2020 and references therein). The

largest uncertainty remains associated to $CH_4$ emissions during the extraction of O&G, with global estimates ranging from 46 to 98 Mt $CH_4$ yr$^{-1}$ (Höglund-Isaksson et al., 2015). This uncertainty is critically related to the diversity of country-specific and site-specific emission factors used. In the US, emissions associated to O&G production where underestimated by a factor 2 (Alvarez et al., 2016), mainly linked to underestimation in upstream (production) emissions. Furthermore, the spatial and temporal distribution of sources is poorly known. In the Barnett Shale area (USA), Zavala-Araiza et al. (2015) found that 10%

of the O&G facilities accounted for 90% of the emissions. Assessment of regional emissions distribution can be provided by mobile measurements targeting facilities.

$CO_2$ emissions linked to O&G production and use represent 53% of total $CO_2$ emissions over the period 2008-2017 (Le Quéré et al., 2018; not including land use change). Unlike $CH_4$, $CO_2$ emissions are to a large extent induced by fuel usage for energy





consumption rather than production. In the near future, $CO_2$ emissions are expected to continue increasing, driven by a strong
demand in Asia (Jackson et al., 2019).

A large fraction of global O&G extraction occurs in the Eastern Mediterranean and Middle East region (EMME). Middle East is the main crude oil production region (32% of world total) with 24.16 million barrel per day, and 17,5% of global natural gas production with 701.12 billion standard cubic meters produced in 2019 (OPEC, 2020). The main O&G fields are located in the Arabian Gulf and neighboring countries, as well as in the Gulf of Suez and the Nile Delta. Gas fields have been recently
discovered in the Levantine Sea. $CH_4$ emissions reported for countries neighboring the Mediterranean Sea and Middle East, amount respectively to 15.53 Mt $CH_4$ yr$^{-1}$ (6.0% of global emissions), and to 21.48 Mt $CH_4$ yr$^{-1}$ (8.3% of global emissions) (Janssens-Maenhout et al., 2017). Increasing $CH_4$ emissions in the Middle East and Africa have been proposed as a contribution to the post-2007 CH4 increase (McNorton et al., 2018).

According to the inventory data compiled by Friedlingstein et al., (2019), Mediterranean countries and Middle East $CO_2$
emissions represent respectively 6.8% and 5.6% of global emissions. $CO_2$ emissions for EMME have increased from 662 MtC yr$^{-1}$ to 831 MtC yr$^{-1}$ over the period 2009-2018, essentially driven by fossil fuel use in Middle-Eastern countries. This represent an annual growth rate of 2.5%, significantly higher than the 1.3% yr$^{-1}$ increase rate of global emissions. $CO_2$ emissions from the Middle East are mostly arising from domestic consumption, while the emission transfers linked to international trade are negligible (-2±7 MtC yr$^{-1}$; Peters et al., 2012). However, inventories may underestimate by a factor 2 emissions from the main
urban centers of the Middle East (Yang et al., 2020).

The EMME region is a transitional zone between mid-latitude climates and subtropical area, located in the high-pressure subtropical ridge (Lelieveld et al., 2012). The northern part of EMME is under a westerlies regime with Eastern Mediterranean located in the outflow from European airmasses in the lower troposphere (Lelieveld et al., 2002), while the southern part experiences trade winds. The EMME region includes a large extent of deserts as well as densely populated areas. Despite the
region's important contribution to anthropogenic GHG emission, very few atmospheric measurements of the distribution of GHG are available in the area (Ricaud et al., 2018), limiting the possibility to reduce uncertainties on regional emissions rates. The Middle East remains undersampled, especially by the in-situ surface networks (Ciais et al., 2014). It also offers fewer cloudy days than other mid-latitude locations, thus enhancing the potential of satellite passive measurements (Yang et al., 2020).

What are the typical $CO_2$ and $CH_4$ mixing ratios around the Arabian peninsula? What are the main drivers for the observed variability and can we link this variability to sources? To what extent can we confirm or inform the inventories based on the measurements, and more specifically the O&G component of these inventories? This paper aims at better understanding the drivers of variability for these species in the area, and to relate this variability to regional sources.

To address these questions, $CO_2$ and $CH_4$ mixing ratio measurements were performed during a ship cruise across the
Mediterranean Sea, Red Sea and Arabian Gulf in summer 2017. Large-scale, ship-borne measurements of $CH_4$ and $CO_2$ atmospheric concentrations have been previously performed for a wide variety of purpose including measuring emission from



O&G platforms (see e.g. Yacovitch et al., 2020), assessing nation-wide emissions in the UK and Ireland (Helfter et al., 2019) or disentangling regional sources in the Arctic (Berchet et al. 2020).

Here we build on the work of Bourtsoukidis et al. (2019) who characterized light alkanes in relation to the various hydrocarbon

sources in the region. We compare these results to a simulation of the anthropogenic component of $CO_2$ and $CH_4$ excess during the cruise using a Lagrangian particle dispersion model.

Section 2 details the AQABA (Air Quality and climate change in the Arabian Basin) campaign, as well as the measurement and modelling methodology. Section 3 presents measurements data and discusses how these data compare with simulations of $CH_4$ and $CO_2$ enhancements linked to anthropogenic activities.

**2. Methods**

**2.1. Campaign and Platform**

The AQABA campaign took place in summer 2017 from June 24 to September 3. The scientific purpose of the campaign was to investigate the atmospheric composition and chemical processes over the Arabian Basin (Pfannerstill et al., 2019; Wang et al., 2020; Bourtsoukidis et al., 2019; Bourtsoukidis et al., 2020; Tadic et al., 2020; Celik et al., 2020). The ship departed from,

and returned to La Seyne sur Mer, near Toulon, France. Fig. 1 shows the ship's route and calling ports. The ship traveled through the Suez Canal and the Red Sea, then around the Arabian Peninsula and through the Persian Gulf (also named Arabian Gulf) to Kuwait, where it anchored for 3 days (from July 31 to August 3) at the port. The ship eventually returned and reached La Seyne sur Mer by approximately the same route on September 2. Several calls occurred at various ports on the way. The ship used was the Kommandor Iona, a UK-based, 76-m long dynamic positioning research and survey vessel. The mean speed

of the vessel during the campaign was $3.4\pm1.8$ m s$^{-1}$.





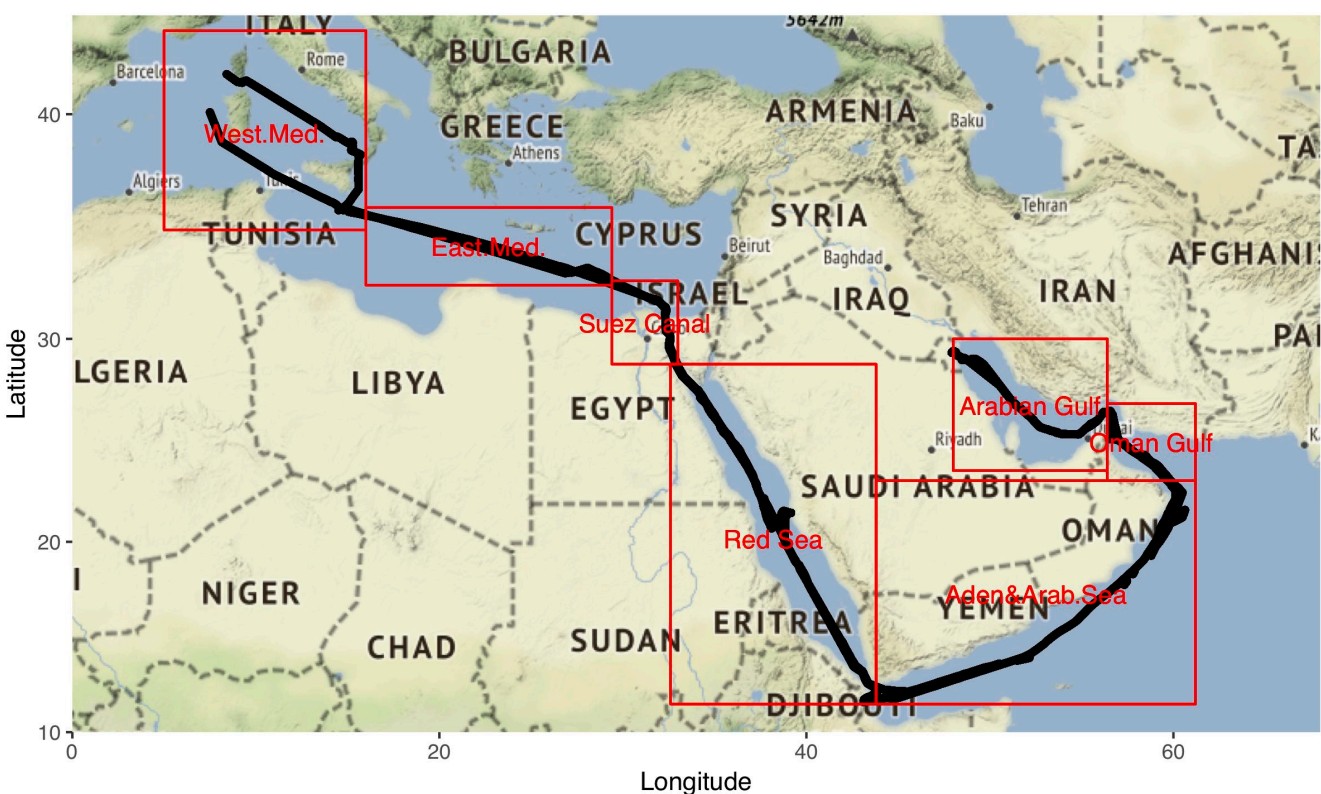

**Figure 1. Ship track during the campaign. Regional limits are indicated. Background map © OpenStreetMap contributors 2021. Distributed under a Creative Commons BY-SA License.**

### 2.2. Measurements

To minimize contamination of measurements by the ship's own emissions, the measurement inlet for $CO_2$ and $CH_4$ was located at the front of the ship, 15 m above sea level. Air was drawn through 15 m of Synflex tubing using a 2 L/min flush pump to prevent degradation of the system response time. The $CO_2$ and $CH_4$ mole fractions were measured in situ using a cavity ring-down spectroscopy analyzer (Picarro G2401; Santa Clara, USA) located in an air-conditioned container on the front deck. CO measurements were also collected. The air was not dried prior to analysis and water vapor effects on $CO_2$ and $CH_4$ were

corrected (Hazan et al., 2016). A filter and a peristaltic pump ensured that no dust or liquid water enter the analyzer. Four different calibration cylinders of compressed air bracketing typical ambient concentrations were injected in the analyzer before departure, every 15 days during the trip, and upon return to Toulon for calibration and quality assurance. Calibration gas concentrations ranged between $377.630\pm0.013$ ppm and $474.845\pm0.013$ ppm for $CO_2$, between $1791.840\pm0.101$ ppb and $2791.010+0.230$ for $CH_4$ and between $126.795\pm0.630$ ppb and $346.120\pm0.609$ ppb for CO. The injection sequence consisted of

four 20-min injections of each of the four gases. An additional target gas of ambient concentration was injected twice daily to



assess measurement accuracy. Precision of the G2401 (expressed as continuous measurement repeatability) is typically better than 0.03 ppm $CO_2$, 0.3 ppb $CH_4$ and 8 ppb CO (Yver-Kwok et al., 2015). The data has been processed and quality-controlled following ICOS (Integrated Carbon Observing System) standard procedure (Hazan et al., 2016), including the propagation of the calibration and threshold-based filters. The concentrations measured from calibration cylinders between two calibration

sequences show a mean drift of 0.05 ppm and 0.5 ppb $CH_4$, significantly below the drifts typically observed at fixed observatories (Hazan et al., 2016). Target injections showed a small residual bias (after calibration) below 0.05 ppm $CO_2$ and 0.3 ppb $CH_4$. The measured target values for CO vary within ±10 ppb sd. The processed data is reported as 1-min averages. Alkanes were measured using Gas Chromatography-Flame Ionization Detector (GC-FID; Bourtsoukidis et al., 2019). GC-FID sampling intervals were 10-30 minutes, with uncertainties ranging between 5 and 10%. Detailed description of the GC-FID

measurements can be found in Bourtsoukidis et al. (2019).

Meteorological parameters such as wind speed and direction, pressure and temperature as well as GPS position and course, were acquired from a meteorological station (Shipborne European Common Automatic Weather Station, EUCAWS) at the starboard side of the front deck of the ship.

## 2.3. Stack contamination data filter

As the sampling inlet was situated at the front of the boat, measurements can be occasionally influenced by the ship's stack emissions. We assume that this influence is mainly depending on the relative wind direction, where a tailwind is likely to bring smoke from the boat's chimney back onto the instrument and thus contaminate the measurements. We flagged 1-min measured concentrations using a binary index indicated possible presence/absence of stack contamination as in Tadic et al. (2019). Different values of angle sector around the stern were tested to filter data using relative wind direction and speed. 16% of data

were flagged as potentially contaminated by the ship and not considered in the following analysis.

## 2.4. Lagrangian modelling

Atmospheric transport was investigated using the FLEXible PARTicle dispersion model (Flexpart) v9 Lagrangian particle dispersion model. Flexpart calculates the trajectories of a large number of tracer particles using the mean winds interpolated from the analysis fields plus random motions representing turbulence (Stohl and Thomson, 1999). Results presented here use

ECMWF analysis fields at 1° resolution. Here, the backward method is used to analyse transport pathways from source regions to the receptor position. Each simulation consists of 10,000 particles released every hour. Released particles were followed up to 14 days backward in time. Potential emission sensitivity is considered when particles reside below the boundary layer height as retrieved from the ECMWF analysis. Potential emission sensitivities are then convolved with gridded surface fluxes from emission inventories to simulate mixing ratios at the ship position during the cruise.



## 2.5. Emission inventories

For simplicity, our simulation uses surface emission of $CO_2$ and $CH_4$ that include only anthropogenic emissions. Data are taken from the Emission Database for Global Atmospheric Research (EDGAR) v4.3.2 for the year 2012 (Janssens-Maenhout et al., 2017; Janssens-Maenhout et al., 2019). EDGAR estimates country total emissions for a variety of species including $CO_2$ and $CH_4$ based on international emission factors and activity data. Maps are eventually generated at a resolution of 0.1°x0.1° according to spatial proxies. For fossil fuels exploitation for examples, spatial proxies combine observed flaring and cartography of infrastructures. Our Flexpart simulations calculate contributions detailed by sector and by country following the typology of the EDGAR database.

The Evaluating the Climate and Air Quality Impacts of Short-Lived Pollutants (ECLIPSE) baseline version CLE 5a (Höglund-Isaksson, 2012) is also used for comparative simulations of $CH_4$ using Flexpart ($CO_2$ is not provided as gridded data). ECLIPSE derives from the GAINS model used for scenario analysis. ECLIPSE maps emissions a resolution of 0.5°x0.5° according to spatial proxies different from EDGAR.

## 3. Results and discussion

### 3.1. Spatial Distribution of $CH_4$ and $CO_2$ along the cruise

Figure 2 shows the concentrations observed during the two legs of the cruise. GPS data acquisition was missing in part of the Western Mediterranean and therefore part of the data is not shown on the map. Figure 3a shows the measured $CH_4$ mixing ratios statistics for each region during the cruise, excluding data influenced by the ship's own exhaust and data acquired when the ship is at rest. Median values in the Western and Eastern Mediterranean were respectively 1903 ppb (interquartile range (IQR) 1896-1907 ppb) and 1906 ppb (IQR 1879-1929 ppb), but significant differences were observed, for each region, between the two legs. For Eastern Mediterranean, Leg 1 median $CH_4$ concentration is 1878 ppb, under a general northern wind influence whereas Leg 2 median is 1929 ppb with a generally southern wind. Closer to the Suez Canal, wind has been more generally from the South. The Gulf of Aden, the Arabian Sea and the Gulf of Oman were markedly lower in $CH_4$, suggesting a different airmass origin. This is confirmed by a back-trajectory analysis using the Flexpart model, showing a markedly different regional pattern in sensitivity, with particles residence time in the boundary layer over Ethiopia, Somalia, and the surrounding ocean. This is also consistent with the very low total OH reactivity observed in the area during the campaign (Pfannerstill et al., 2019). Wind direction in this area is mostly parallel to ship track.





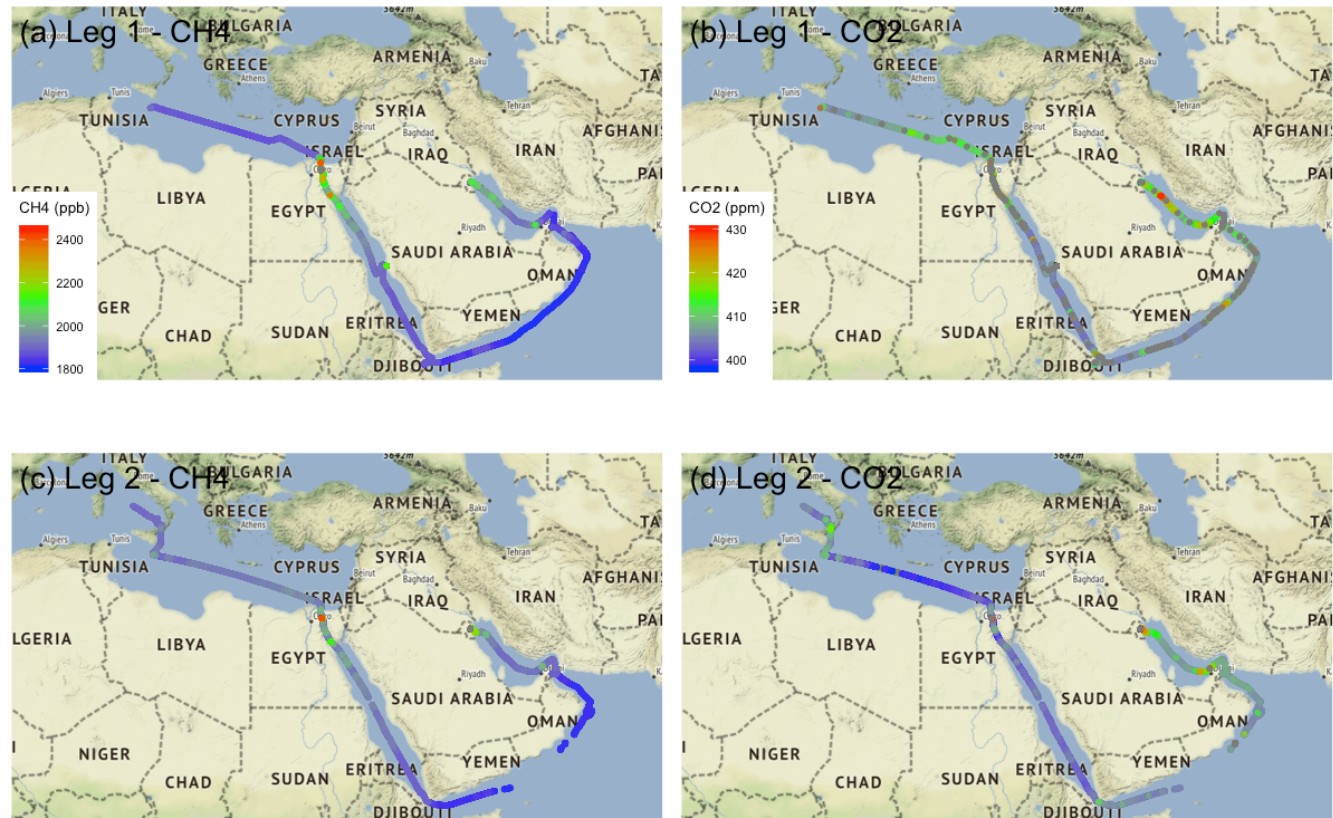

**Figure 2. Measurements of methane (a, c) and carbon dioxide (b, d) collected during Leg 1 (top row) and Leg 2 (bottom row). Background map © OpenStreetMap contributors 2021. Distributed under a Creative Commons BY-SA License.**

Figure 3b shows the $CO_2$ mixing ratio statistics for each region and each leg observed during the campaign. For each area, $CO_2$ concentrations were higher during Leg 1 compared to Leg 2, likely reflecting hemispheric-scale drawdown in $CO_2$ associated to uptake by Northern Hemisphere vegetation (see e.g. Ramonet et al., 2010). The highest median concentrations are observed in the Arabian Gulf (410.27 ppm, IQR 407.16-415.36 ppm) whereas the lowest concentrations were encountered in August over the Eastern Mediterranean, with 403.30 ppm (IQR 399.12- 407.64 ppm), the closest value to the Mace Head mean mixing ratio in July and August. Ramonet et al. (2010) proposed that the minor changes in concentrations differences observed over Eastern Europe compared to the marine baseline are due to biospheric uptake in Europe being offset by anthropogenic (fossil fuel) emissions along airmass pathways. Here, airmass origin is over Eastern Europe's boundary layer over the previous 10 days according to Flexpart backtrajectories.

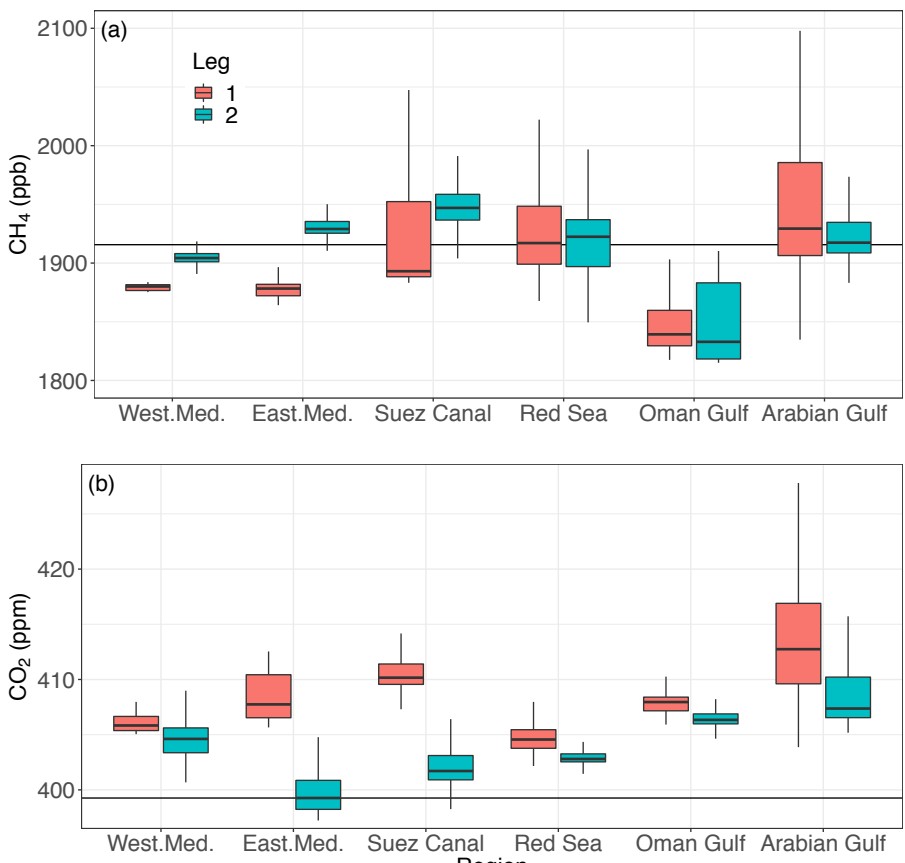

**Figure 3. Methane (a) and carbon dioxide (b) mixing ratios observed during the cruise. Data from ship at rest or contaminated by ship own exhaust are excluded. The box plot shows the interquartile range (25-75%) and median values for each grouping of data. The bars extend to the lowest and highest values, (ignoring outliers beyond 1.5 times the interquartile range). The statistics of the two legs are represented separately. The horizontal black line is the mean methane mixing ratio at Mace Head, Ireland (NOAA) during July and August 2017. The median mixing ratio of $CO_2$ over the Aden and Arabian Sea are comparable to Red Sea and Mediterranean values, contrary to $CH_4$ where mixing ratios were distinctly lower. $CO_2$ mixing ratios do however exhibit a much lower variability (interquartile range) compared to other regions.**

In the remainder of this paper, enhancements of $CH_4$ and $CO_2$ are defined as excess concentrations over a background that is defined for each region and each leg, according to Eq. 1.

$$\Delta X = [X] - [X]_{background} \qquad (Eq. 1)$$

The background is defined for the whole campaign as the tenth percentile of measured mole fractions.





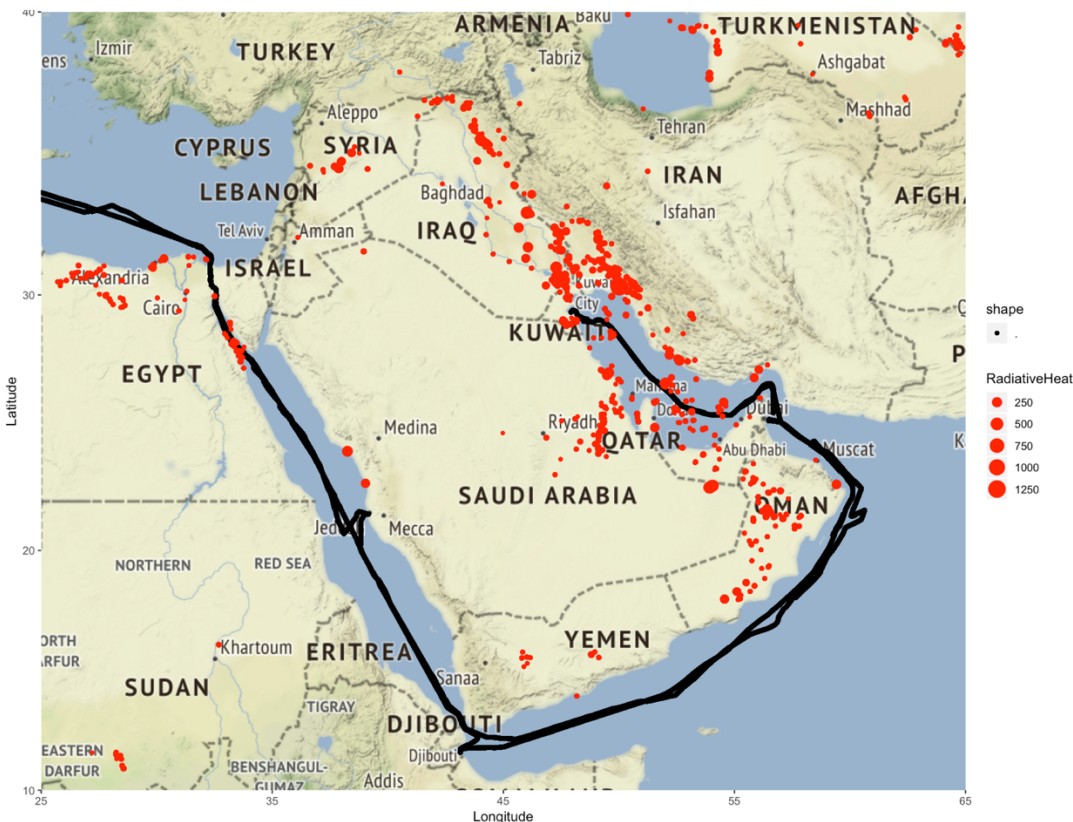

**Figure 4. Ship track superimposed with NASA/NOAA Visible Infrared Imaging Radiometer Suite detected flaring sources active in 2017. Dot size is proportional to the highest observed radiative heat, which in turn is related to the volume of gas flared (Elvidge et al., 2016). Flaring occurs mostly at the upstream production sites, and is therefore a proxy for the presence of extraction and production sites. Flaring spots retrieved from skytruth.org. Background map © OpenStreetMap contributors 2021. Distributed under a Creative Commons BY-SA License.**

### 3.2. Significance of fugitive sources based on light alkanes measurements

In this section we investigate the origin of $CH_4$ enhancements in the regions where the highest median mixing ratio were observed. These regions, the Arabian Gulf, the Suez Canal and the Red Sea, are all areas where O&G exploration is important, both offshore and onshore (Fig. 4). Although the ship's track did not target specifically offshore platforms, the cruise collected a significant amount of strong $CH_4$ enhancements coinciding with crossing of offshore platforms. Bourtsoukidis et al. (2019) showed that emissions from O&G were significant contributions to non-methane hydrocarbons measured during the cruise over several areas around the Arabian Peninsula. According to the EDGAR emission inventory (Janssen-Maenhout et al., 2017), the O&G industry is responsible for up to 80% of $CH_4$ emissions in the Middle East. Local enhancements of mixing ratio in airmasses are typically linked to local or regional sources. The most likely origin of the high $CH_4$ and $CO_2$ concentrations over the Arabian Gulf and the Suez Canal is anticipated to be anthropogenic emissions. In this section we investigate the origin of $CH_4$ enhancements by assessing their correlation with non-methane hydrocarbons (NMHC) measured onboard the ship.





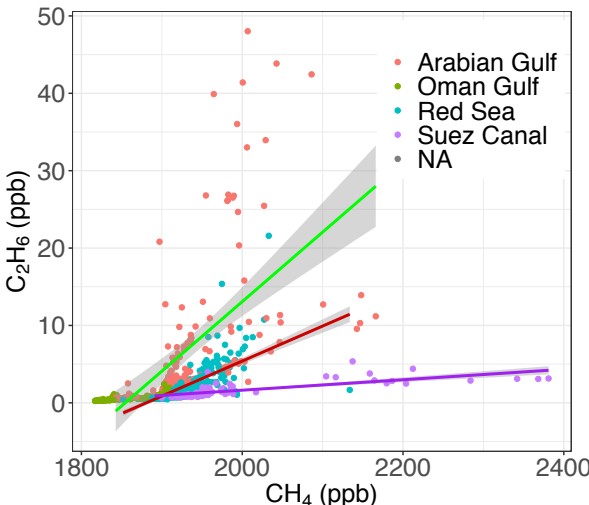

**Figure 5. Scatter plot of ethane and methane for the Gulf of Oman gulf, Arabian Gulf, Red Sea and Suez Canal. The regression line and associated 95%-confidence interval are plotted for each of these 4 regions.**

Over the whole campaign, we observed a significant but small correlation between $CH_4$ and $C_2H_6$ ($r = 0.386$, $p < 10^{-15}$). The

highest $C_2H_6$ concentrations are observed over the Arabian Gulf (Fig. 5). In this region, a significant correlation between $C_2H_6$ and $CH_4$ ($r = 0.332$, $p = 0.004$) was derived, with a region-wide regression slope of 0.062 ppb ppb$^{-1}$. Similarly, $CH_4$ was correlated with propane ($r = 0.356$, $p = 0.002$) and with the sum of $i$-butane and $n$-butane ($r = 0.360$, $p = 0.002$). $CO_2$ showed no significant correlation with any of these three NMHCs. This suggests that $CH_4$ enhancements in the area are at least partly explained by fugitive O&G sources and are not related to combustion sources.

Over the Red Sea, $CH_4$ enhancements of about 200 ppb above background have been observed. A clear correlation between $C_2H_6$ and $CH_4$ ($r = 0.71$, $p < 10^{-15}$) is associated with a regression slope of 0.047 ppb ppb$^{-1}$. For the Suez Canal area, the regression slope between $C_2H_6$ and $CH_4$ is 0.006 ppb ppb$^{-1}$ ($r = 0.75$, $p < 10^{-15}$). For the Suez Canal the airmass originates from the Eastern Mediterranean and Egypt according to Flexpart backtrajectory simulations. Differences in regional $C_2H_6$ to $CH_4$ regression slopes may therefore reflect either an aggregation of different O&G emission sources or different fraction of O&G

in observed $CH_4$ enhancements.



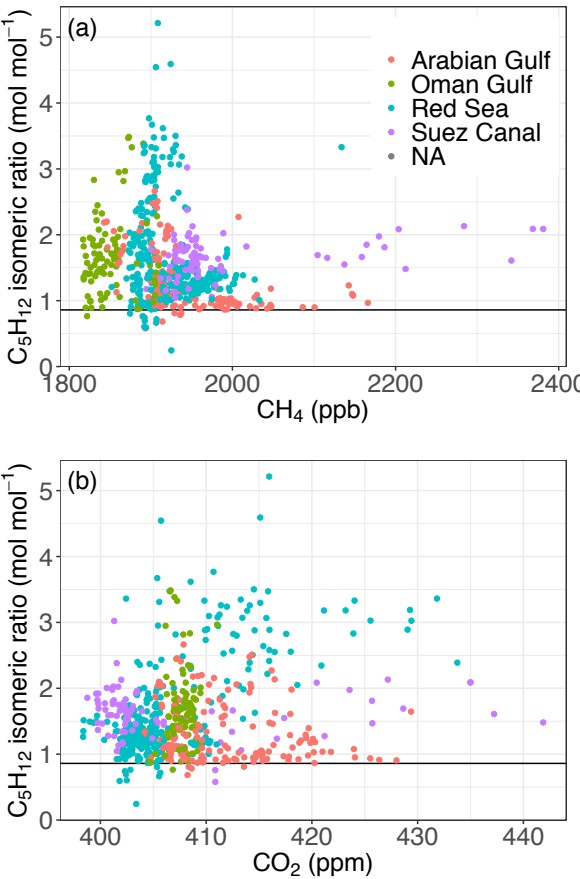

**Figure 6. Scatter plot of i-pentane/n-pentane and methane (a) and i-pentane/n-pentane and CO₂ (b) for Oman Gulf, Arabian Gulf, Red Sea and Suez Canal. The value of i-Pentane to n-Pentane ratio associated with oil and gas emission (0.86) is drawn as a horizontal line in the two panels.**

Ethane is a valuable proxy for fossil fuel emissions at the global scale and has provided evidence of the reduction of fossil fuel emission in the 2000s (Simpson et al., 2012). As a proxy at the site scale it contributes to the separation of thermogenic and biogenic emissions (e.g. Rella et al., 2015; Assan et al., 2017; Lowry et al., 2020, Defratyka et al., 2020). However, the range of $C_2H_6:CH_4$ ratio can be highly variable at regional scale and from emission site to site. Measuring downwind of >100 offshore platforms in the Gulf of Mexico, Yacovitch et al. (2020) found $C_2H_6:CH_4$ ratio ranging from 0.16−17%. The *Deepwater*

*Horizon* oil spill associated gas sampled in the hydrocarbon plume rising from the seafloor showed $C_2H_6:CH_4$ ratio between 8.1% and 8.3% (Reddy et al., 2012). This ratio depends on the temperature of formation of natural gas (Whitticar, 1994). The ratio can be further modified if a significant fraction of $CH_4$ is emitted through flaring, with different depletion during combustion (Yacovitch et al., 2020). Hence due to its inherent variability from field to field, the $C_2H_6:CH_4$ ratio alone is insufficient without additional constraints to quantify the relative contribution of O&G to $CH_4$ emissions at the regional scale.





Figure 6 shows the *i*-pentane to *n*-pentane ratio for selected regions. This ratio is another tracer of the contribution of natural gas, vehicle exhaust or fuel evaporation emissions to the atmospheric burden of alkanes and associated species (Gilman et al., 2013; Bourtsoukidis et al., 2019). For natural gas the ratio is expected to be 0.86 (indicated as horizontal line in Fig. 6). Since the ratio is conserved against oxidation by OH in the atmosphere the tracer is conserved during atmospheric transport. In the Arabian Gulf, where the mean $CH_4$ enhancement was 67 ppb and the maximum 255 ppb, the linear regression slope was found

to be 0.939±0.023ppb ppb$^{-1}$. The mean ratio of the isomers was 1.00±0.12, suggesting a clear dominance of fugitive sources linked to O&G production.

### 3.3. Photochemical age and remoteness of sources

VOCs have different atmospheric lifetimes and, if species are co-emitted at the same time and with a known emission ratio, measured changes in their ratios can help trace the photochemical age of an airmass (Purvis et al., 2003), and hence provide

an indication of a transport time from the source to the receptor. The ratio of species is less sensitive to dilution and air mass mixing that the concentration of individual species. Transport time and photochemical age correlate well in particular for lighter alkanes (Parrish et al., 2007). For the AQABA campaign, Pfannerstill et al. (2020) used decreases in the $[C_3H_6O]/[C_3H_8]$ ratio and simultaneous increases in the toluene∕benzene ratio in the Arabian Gulf as an indicator of nearby sources. Wang et al. (2020) analyzed the mismatch between the toluene∕benzene measured ratio and emission ratio. Here we compare, for

several regions, 1) the decorrelation that occurs between $CH_4$ and light alkanes (propane and pentane) during early photochemical ageing due to differences in kinetic rates against OH and 2) the photochemical age of airmasses exhibiting significant correlations between propane and pentane.

Propane ($C_3H_8$) has a lifetime of ca 5.5 days while pentane ($C_5H_{12}$) has a lifetime of 0.5-1 days with respect to OH radical oxidation (Bourtsoukidis et al., 2019). Fig. 7 shows scatter plots of $CH_4$ to propane and pentane (the latter as the sum of n-

pentane and i-pentane). Correlation between $CH_4$ and propane is relatively homogeneous across regions albeit with very different slopes. For the Arabian Gulf the correlation coefficient is $r = 0.37$, for the Gulf of Oman $r = 0.46$, whereas for the Suez Canal we obtain $r = 0.50$. The correlation of $CH_4$ to pentane is more contrasted. For the Arabian Gulf and Suez Canal the correlation coefficient remain significantly high ($r = 0.40$ and $r = 0.67$ respectively), which is comparable to the propane- $CH_4$ correlations. The significant correlations suggest that the airmass is affected by emissions with constant ratios, and hence there

is a regional signature of local sources. This corroborates the results of Wang et al. (2020) finding limited departure between measured ratio and emission ratio for these two regions. In contrast, in the Gulf of Oman there is no correlation between $CH_4$ and propane ($r = 0.003$). This suggests that high $CH_4$ concentrations observed in the Arabian Gulf and Suez Canal are mainly determined by relatively nearby sources co-emitting $CH_4$, pentane and propane. This approach suggests that excess $CH_4$ observed over the Gulf of Oman is emitted from remote sources or without co-emission of these species.

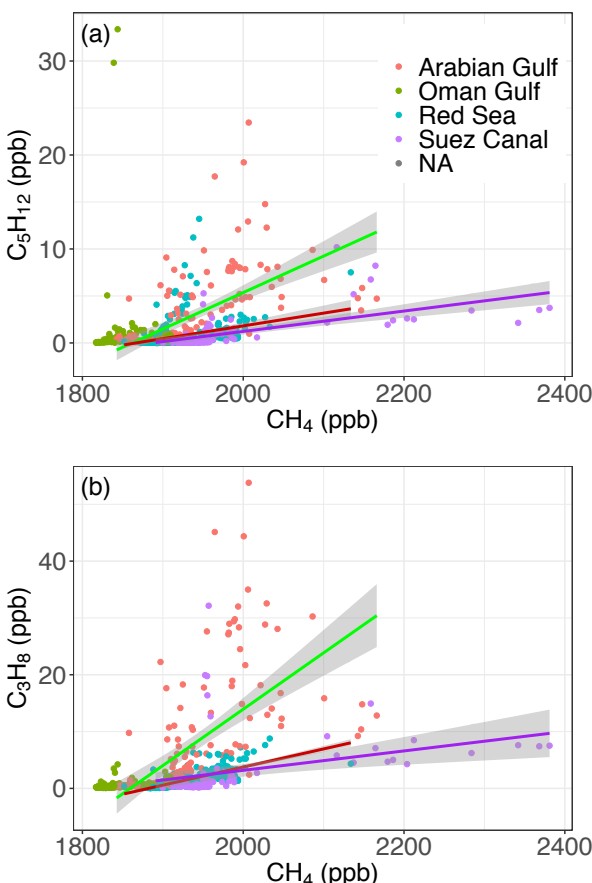

**Figure 7. Scatter plot of (a) pentane and (b) propane against methane for the Gulf of Oman, Arabian Gulf, Red Sea and Suez Canal. The regression line and associated uncertainty are plotted for each of these 4 regions.**

As a next step we evaluate the average photochemical age of airmasses for the Suez Canal and Arabian Gulf regions. For that purpose, we examine, at the regional scale, the change in measured propane to (n + i)-pentane ratio relative to the initial emission ratio. We follow the definition of photochemical age against OH defined in Parrish et al. (2007), given here in Eq. 2 below.

$$t_a \langle [OH] \rangle = \int_{t=t_E}^{t_R} [OH]\, dt = -\frac{1}{\langle k_A - k_B \rangle} \left\{ \ln\left(\frac{[A]}{[B]}\right) - \ln\left(\frac{[A_0]}{[B_0]}\right) \right\} \quad \text{Eq.2}$$

Where $t_a$ is the average photochemical age against OH, $\langle [OH] \rangle$ is the mean concentration of OH along pathway, $t_E$ refers to time of emission and $t_R$ time of reception, A and B are the two species propane and pentane, and $A_0$ and $B_0$ are the initial ratio at the time of emission. We evaluate the mean ratio of concentration between species at the regional scale as the regression slope between the two species. The regression slope of pentane against propane is 0.149 mol mol$^{-1}$ for the Suez Canal and



0.378 mol mol⁻¹ for the Arabian Gulf. The ratio of species at the time of origin is approximated by the emission ratio from the EDGAR v4.3.2 inventory for VOCs (Huang et al., 2017). The inventory-based emission ratio is 0.460 mol mol⁻¹ for the total emissions of Middle East countries and 0.435 mol mol⁻¹ for Egypt. Overall this value varies within a limited span (ranging from 0.347 mol mol⁻¹ for Kuwait to 0.682 mol mol⁻¹ for Iran) across countries with significant emissions of these two species. We use kinetic rate coefficients as reported by Pfannerstill et al. (2019, their Table S1) and assume a mean OH concentration of $7.5\times10^6$ molecules cm⁻³ following Wang et al. (2020). This approach yields a photochemical age for our airmasses of 1.5 days for the Suez Canal and 0.28 days for the Arabian Gulf. At a median true wind speed of 3.9 m s⁻¹ (as observed over the Arabian Gulf and over the Suez Canal), this suggests that sources are distant on average by 206 km and 38 km respectively for the Suez Canal and the Arabian Gulf.

However, this crude approach to the assessment of remoteness of sources of $CH_4$ excesses in the Arabian Gulf and the Red Sea is not sufficient on its own and should be compared to tagged transport simulations.

**3.4. Tagged tracer simulation of anthropogenic methane**

We simulated the anthropogenic excess mixing ratio of $CH_4$ and $CO_2$ using Flexpart and the EDGAR emission inventory. Fig. 8 shows a comparison of simulated excess $CH_4$ time series at the receptor position, and $\Delta CH_4$ measured from the ship. The model reproduces reasonably well the region-wise variability of the signal. In the distinctly low $CH_4$ concentration area over the Gulf of Aden and Arabian Sea the model appropriately simulates enhancements close to zero. In the high concentration area of the Arabian Gulf (from 28 July to 6 August) the model captures the presence of strong enhancements, albeit poorly positioned relative to the known source locations. Over the Red Sea, the model captures the variability correctly. The regional baseline of the signal is however challenging to reproduce with the model since measurements are expressed as excess $CH_4$ with a background calculated by leg and across all regions. This highlights sharply the difference between the low concentration areas (Gulf of Aden and Arabian Sea) versus the other areas more influenced by anthropogenic emissions.

The O&G emission contribute overall to 35% of the simulated excess $CH_4$. Over the Arabian and Oman Gulfs, it accounts respectively for 81% and 63% of $CH_4$ excess on average, consistent with the regional emission pattern and alkane measurements presented above. Over the Red Sea, O&G emission of $CH_4$ explains 37% of the simulated variability on average. Waste management represents on average 27% of the simulated $CH_4$ excess for the whole campaign. In the Arabian Gulf, waste management represents only 12% of $CH_4$ variability.







**Figure 8. Comparison of simulated excess CH₄ time series at the receptor position, and hourly average ΔCH₄ measured from the ship (black line). The simulation combines Flexpart potential emission sensitivity and EDGAR inventory (stacked areas). The main source sectors are indicated as color code. The potential emission sensitivity is also combined with the Eclipse inventory for comparison (blue line). The stops at harbor are indicated as shaded grey areas, with the names of harbors indicated at the top.**

335    The observations at harbor locations (grey shading in Fig. 8) exhibit strong diurnal pattern well reproduced by the model, probably due to the proximity of the land mass and the diurnal cycle of the atmospheric boundary layer development, although the amplitude of the diurnal cycle is rather underestimated by the model, notably for the stop-over in Jeddah between 10-13 July). At harbors, the O&G signal remains dominant, especially in Kuwait. However, the simulated share of waste management sources to the total CH₄ enhancement is twice as much compared to that found over open sea.

340    The highest CH₄ concentrations over the Arabian Gulf are observed mostly in its northern part while the simulation predicts higher concentrations in the southern part, especially in the vicinity of the South Pars/North dome gas field. However, the CH₄ enhancement observed in the immediate vicinity of this gas field is much lower than the simulated one. The model points to an 'O&G exploitation' source type in the Northwestern half of the Arabian Gulf for the strongest CH₄ enhancements observed. Closer to Kuwait, retroplumes show a stronger sensitivity to potential sources in Iran.

345





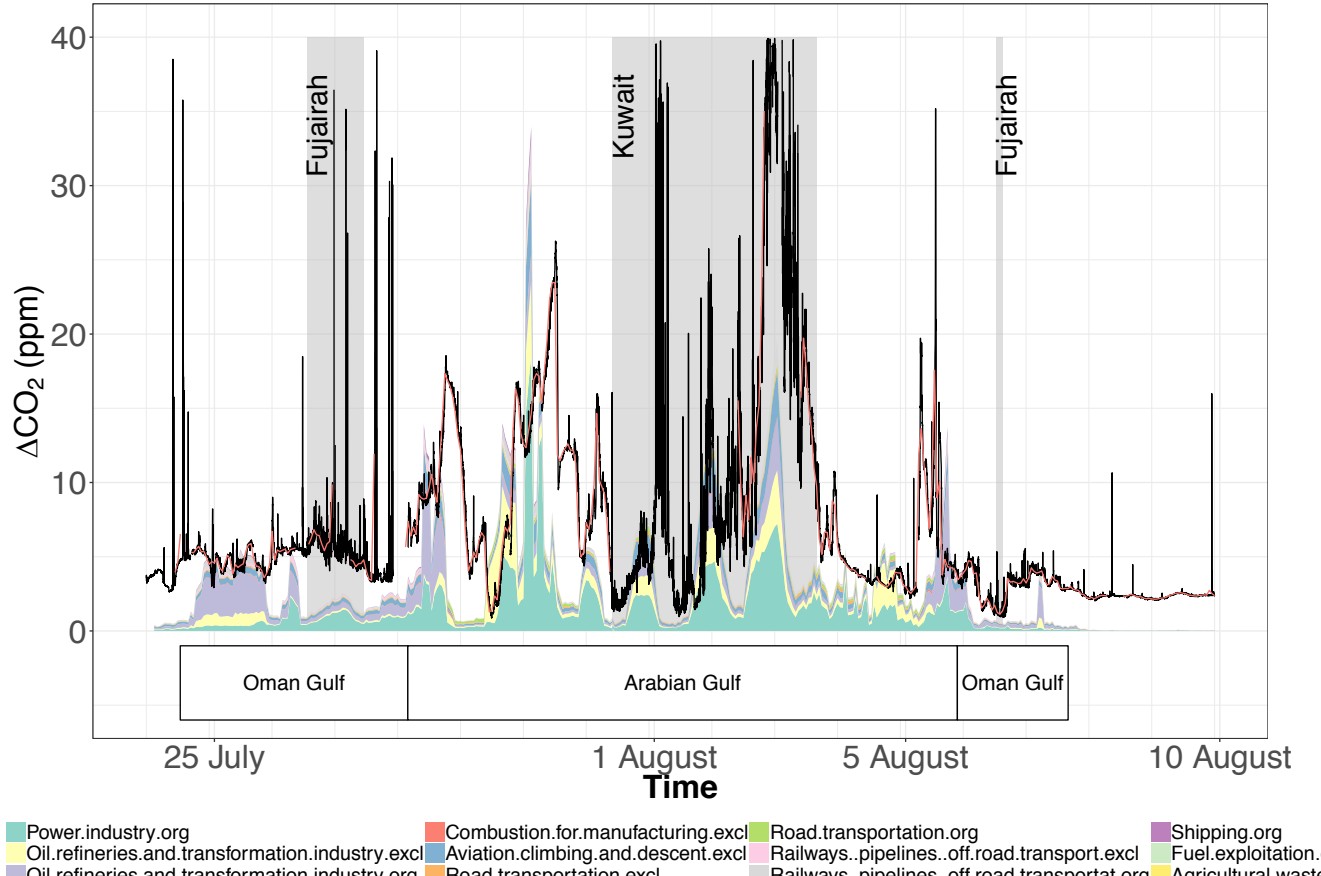

**Figure 9. Comparison of simulated excess $CO_2$ time series at the receptor position in the Gulf of Oman and Arabian Gulf, and hourly average $\Delta CO_2$ measured from the ship (black line). Flagged, hourly average data is shown in light red. The simulation combines Flexpart potential emission sensitivity and EDGAR inventory (stacked areas). The main source sectors are indicated as color code. The stops at harbor are indicated as shaded grey areas, with the names of harbors indicated at the top.**

Figure 9 shows the measured excess $CO_2$ while navigating in the Gulf of Oman and the Arabian Gulf. The simulated excess is dominated by the power industry sector accounting for most of the variability, with contribution from the oil refineries and oil transformation sectors, with local maxima off the coast of United Arab Emirates. This area hosts the Jebel Ali Oil refinery dedicated to the production of liquefied petroleum gas, naphta and a variety of fuel types as well as, further South, the Takreer Abu Dhabi Oil Refinery (retrieved from https://www.industryabout.com/arabian-peninsula-industrial-map).

Figure 10 shows the contribution of countries or areas of origin to simulated $CH_4$ in the Arabian Gulf during Leg 1, where emissions are dominated by fossil fuel exploitation (see discussion on alkanes above and simulation in Fig. 8). By tagging the simulated $CH_4$ enhancements by source region, it is possible to link the contributions of source regions to the ability of the model to reproduce the observed signal.





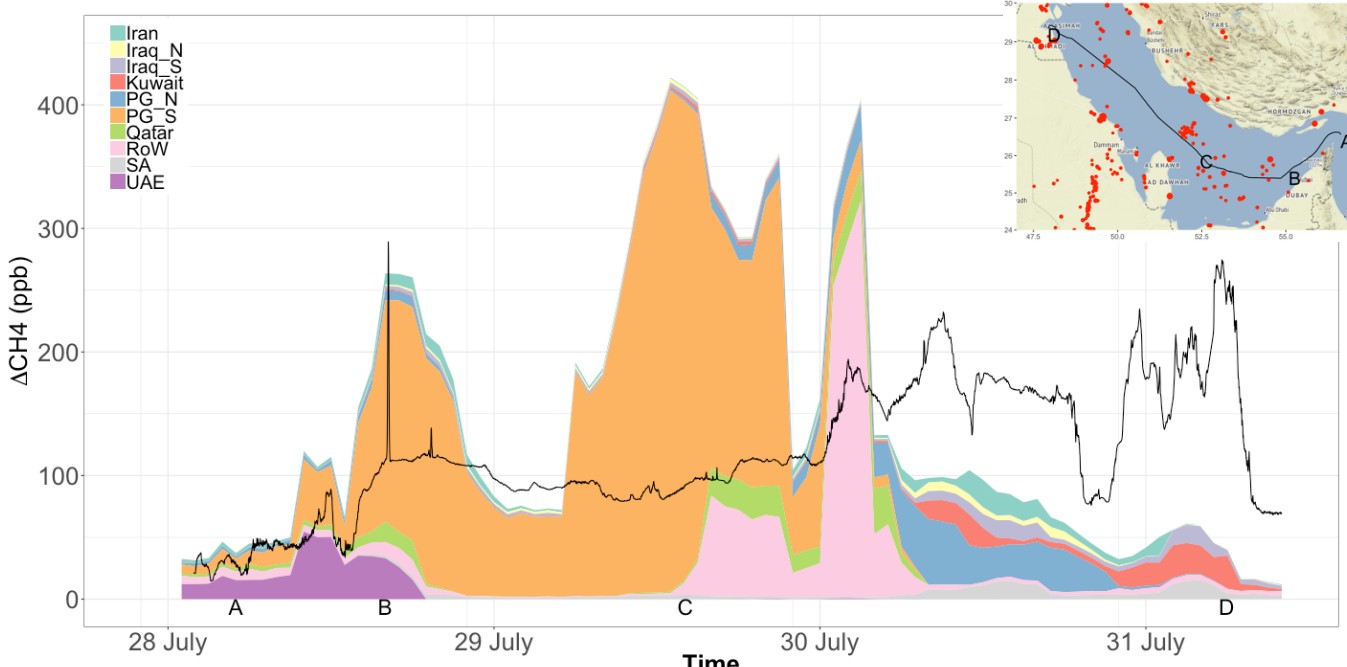

**Figure 10. Simulation of ΔCH₄ by country of origin over the Arabian Gulf. The solid line shows the measured DCH₄. The simulated excess CH₄ is tagged and color-coded by country or area of origin. Iraq_N : Northern Iraq, Iraq_S: southern Iraq; PG_N: Persian (Arabian) Gulf, Northern half; PG_S: Persian (Arabian) Gulf, Southern half; RoW: Rest of the World; SA: Saudi Arabia; UAE: United Arab Emirates. The inset shows the trajectory of the ship corresponding to the simulation, and the labels A, B, C, D highlight points of the ship's route discussed in the main text. Red points correspond to flaring detected from space (see Fig. 4) at the time of the campaign. Inset background map © OpenStreetMap contributors 2021. Distributed under a Creative Commons BY-SA License.**

Off the coast of Dubai and United Arab Emirates (Transect A-B in Fig. 10), the simulation is in good agreement with the observations, and points to a strong contribution from the nearby state (United Arab Emirates) and the local offshore activities in the Arabian Gulf. Within a thin plume lasting 15 minutes (labeled B in Fig. 10, 18:05-18:20 UTC) the highest CH₄ concentration of the section is measured (2109 ppb). The model successfully captures this enhancement with an average of 150 ppb excess CH₄, albeit spreading it over two hours. It is simulated as coming mostly (80%) from local offshore emissions (Arabian Gulf South in Fig. 10) while the ship is moving through the Fateh oil field off Dubai. Approaching the North field off the coast of Qatar (point C, and during 29 July), however, CH₄ is strongly overestimated by the model, with simulated enhancements of more than 400 ppb having no equivalent pattern in the measurement. This strong enhancement is due, in the model, to local offshore emissions (southern part of the Arabian Gulf) associated to the South Pars/North Dome gas field. South Pars/North Field is the largest gas field in the world, shared between Iran and Qatar (Conti et al., 2016). In the Northern part of the Arabian Gulf (transect from C to D) significant CH₄ enhancements are measured and the model fails at reproducing these enhancements, although it suggests local emissions (Northern Persian Gulf and Kuwait) are dominant, consistent with the photochemical age of airmasses (Sect. 3.3).





## 3.5. Model-data comparison

Can we confirm or verify the inventories based on the measurements? The entire dataset (excluding data flagged as contaminated and stationary measurements at harbors) has a correlation of $r = 0.13$. The agreement improves during nighttime with $r = 0.22$. Overall the simulation represents well the variability of $CH_4$ at the synoptic scale. The first order discrepancies arise from the difference in background between regions, since here the background used to offset the $CH_4$ measurements is calculated for the whole campaign. While the measured $\Delta CH_4$ is an excess over a pre-defined background, the simulated excess $CH_4$ integrate anthropogenic surface sources during 14 days prior to reaching the receptor. Therefore, in order to compare these two quantities, the mismatch between the boundary conditions of the model (i.e. $CH_4$ of the airmass prior to the 14 days before measurement) and the background defined for the measurement data introduces a varying offset. However, the simulated time series can be considered as an indication of the geographical or sectoral origin of the observed excess mixing ratio.

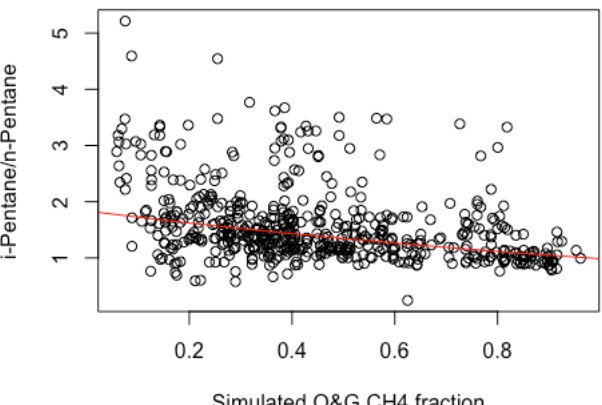

**Figure 11. Scatter plot of the fraction of simulated methane due to O&G sources against i-pentane to n-pentane ratio. An exponential model is fitted to the date (red line).**

Figure 11 provides a comparison of the fractional contribution of O&G exploitation sources to simulated $CH_4$ excess with the $i$- to $n$-pentane ratio (see Sect. 3.2). We find that an exponential decrease model can fit the data with a significant correlation $r = -0.38$ ($p < 10^{-16}$). This exponential model yields an estimated $i$- to $n$-pentane ratio of 0.99 when the simulation fraction of O&G is 100%, whereas absence of O&G exploitation would yield a ratio of 1.84. This confirms that the fraction of emissions linked to O&G exploitation in the model is consistent with our expectations according to the isomeric ratio of pentane (Bourtsoukidis et al., 2019).

The inventory used for the simulation also integrates a large part of uncertainty, in terms of intensity and location of the sources. To test the sensitivity to the emission inventory itself, we simulated $CH_4$ excess using the Eclipse inventory for $CH_4$ (light blue line in Fig. 9). The two inventories agree to a large extent for all regions apart from the Arabian Gulf, where Eclipse



yields a mean excess that is 65% higher than the EDGAR simulation (mean 255 ppb and 156 ppb respectively). For the Arabian Gulf, none of the inventories yield significant correlation, suggesting a poor representation of spatial distribution of emisions. Eclipse also simulates much higher $CH_4$ excess in the vicinity of the Fateh oil field, with a local maximum of 566 ppb, compared with the EDGAR simulation that yields 150 ppb. In the Northern part of the Arabian Gulf, Eclipse also simulates

higher excess concentrations than EDGAR, but the Eclipse simulation shows a much better agreement with the observed $CH_4$ variability, especially at the Kuwait coast on July 30 and 31. The enhanced divergence among simulations using different inventories is most likely due to the increasing differences between inventories at finer spatial scales, as has been highlighted by Ciais et al. (2010) for $CO_2$ in Europe. Furthermore, the high skewness in the regional distribution of emission for each facility may not be properly represented in the inventories (Zavala-Araiza et al. 2015), leading to strong discrepancies between

simulation and measurements when measuring directly near such facilities. Overall, in the Southern part of the Arabian Gulf, where gas fields represent a larger part of the O&G extraction, EDGAR tends to accurately represent specific plumes (see point B in Fig. 10), which is not the case for ECLIPSE. However, on average both EDGAR and ECLIPSE tends to overestimate the signal (Fig. 8). In the Northern part of the Arabian Gulf, where oil extraction is comparatively more represented the EDGAR inventory tend to underestimate the measurements while ECLIPSE underestimate the measurements for Leg 1 and

overestimates it for Leg 2.

Accurately simulating plumes measured in the vicinity of point sources is highly dependent on the injection position of emissions in the inventory. Here we used inventory maps in the model with resolution of 0.1°x0.1° for EDGAR, hence a point source is spread within 10 km. This may result in discrepancies between observations and simulated fields. This is particularly true if an important source point is within the distance corresponding to the spatial resolution of the wind fields driving the

model (here, 1°x1°, i.e. approximately 100 km). This may cause the strong overestimations of $CH_4$ enhancements over the south-eastern part of the Arabian Gulf.

Unaccounted-for time variations may also play a role in model-observation mismatch. Since the inventory is static, any daily or weekly pattern in emissions would not be reproduced in the model, which could be important when the ship is in harbors. Moreover, venting, incomplete flaring combustion and maintenance activities are typically leading to intermittent $CH_4$

emission. In a study of black carbon emissions from flaring in Siberia, Petäjä et al. (2020, their Sect. 3.8.2) found that accounting for actual times of flaring instead of using annual means can significantly improve the simulation of downwind atmospheric measurements.

Since our study is focusing on anthropogenic emissions and given low natural emissions in the Eastern Mediterranean and Middle East area (Saunois et al., 2020) we used only anthropogenic inventories in our study. However, some level of sensitivity

to natural fluxes can be expected. The $CO_2$ sink would affect concentrations downwind of large forested areas, for example around the Eastern Mediterranean basin. But biogenic sinks are not likely to play a role in the desert dominated regions of the Arabian Peninsula.

O&G extraction at the country level varies from year to year, and the life cycle of individual productive fields evolves over several years. As a result, changes over time of the emission spatial pattern and intensity as reported in inventories for specific



years may affect the accuracy of the simulation. Here, our simulation uses the EDGAR inventory for year 2012. Oil production
in the Middle East increased by 10.8% between 2012 and 2017, while gas production increased by 19.2% (Dudley et al., 2019).
According to the most recent available update of the inventory (EDGAR v5.0), O&G emissions have increased by 8.61% for
countries bordering the Arabian Gulf between 2012 and 2015, the latest available year. In order to investigate the potential
evolution of the oil extraction activity the evolution of flaring activity between 2012 and 2017 has been investigated. Flaring

data is obtained from VIIRS Skytruth nightlight product (Elvidge et al., 2016) for the Middle East. Flaring has shown a twofold
increase in intensity (in terms of number of hotspots detected from space) over the period, which suggests that the 8.61%
increase in inventory O&G emissions from the neighboring countries is not sufficient to match the increase in extraction
activity. The spatial distribution of the flaring has not varied significantly over the period. In addition to flaws in the inventory
for its base year, accounting for increased activity over the period 2012-2017 would lead to further overestimations between

model and observations linked to O&G activity in the Arabian Gulf. This therefore does not contribute to explaining the model
overestimation, especially in the Southern part of the Arabian Gulf.

**Conclusion**

The AQABA campaign provided a first overview of the regional distribution of $CH_4$ and $CO_2$. Three distinct elements are
identified in the distribution of GHG during the campaign. The Mediterranean is dominated by European emissions. The Gulf

of Aden and the Arabian Sea are in an air mass relatively poor in GHG, with air masses originating from Eastern Africa. The
Red Sea and the Oman and Arabian Gulfs showed high $CO_2$ and $CH_4$ concentrations.
The $C_2H_6$:$CH_4$ and i-pentane to n-pentane ratios suggest that over the Red Sea, Oman and Arabian Gulfs, $CH_4$ enhancements
are originating from O&G emissions, especially in the latter area. Repeated $CH_4$ enhancements over the Suez Canal, Red Sea,
Arabian Gulf are also unambiguously identified as emitted from local O&G extraction/exploitation (Arabian Gulf). This is

clearly supported by a Lagrangian simulation based on the EDGAR inventory, showing that 81% of excess $CH_4$ over the
Arabian Gulf is due to the exploitation of O&G. The $CO_2$ variability is dominated in this area by anthropogenic emissions,
with a dominant contribution from the power industry and the oil refining and transformation sectors.
While the model predicts a dominant contribution from O&G exploitation to $CH_4$ enhancements, in agreement with the light
alkanes' measurements, only weak quantitative agreement has been found between modelled and measured $CH_4$. This is most

likely explained by a combination of factors including error in the inventory, poor dilution in the model in the vicinity of the
sources, poor distribution of the point source intensities and lack of representation of temporal variations in emission patterns.
A similar simulation using a different inventory (ECLIPSE) tended to overestimate the measured $CH_4$ close to sources in the
Southern Arabian Gulf. Despite these compounded uncertainties our study provides strong indication that inventories
overestimate part of the regional upstream O&G emissions in middle-eastern countries neighboring the Arabian Gulf,

especially linked to gas extraction. On the opposite, in the Northern Arabian Gulf with comparatively more oil fields the
measured methane is generally underestimated by the simulations. Increase in O&G emissions in the Middle East compared



to the reference year of the inventories would further enhance overestimations but could partly explain the underestimation for the Northern Arabian Gulf area. Our study also shows that the inventories must be improved for the spatial distribution of emissions in this area whose emissions are dominated by the sector of O&G extraction, transport and transformation. More targeted measurements investigating specifically offshore and onshore extraction sites will enable a better understanding of the distribution of emission in the area. Combining $CH_4$ and alkane measurements offers the potential to separate the O&G component of emitted $CH_4$, to investigate at the regional level the fraction of emissions linked to O&G exploitation.

**Data availability**

The data is available on reasonable request directly from the authors.

**Authors contribution**

JDP conceived this study and wrote the manuscript. AR and AB performed and analysed the Flexpart simulations. EB, JW and LE did the alkane measurements. JDP and MD did the greenhouse gas measurements. IT provided complementary data. HH was the chief of mission. JL was the PI of the campaign. All contributed to the manuscript.

**Competing interests**

The authors declare that they have no conflict of interest.

**Acknowledgements**

Max Planck Institute for Chemie, Mainz, organized the AQABA campaign and chartered the ship. Thanks to Marcel Dorf, Dieter Scharffe, Charlotte Beall, Claus Koeppel, Jean Sciare and many other participants to the cruise for their crucial contribution. Olivier Laurent, Céline Lett and Laurence Vialettes (ICOS teams at LSCE) were key contributors in designing and implementing the measurement setup. Thanks to Hays Ships and the R/V Kommandor Iona crew for their successful and friendly implementation of the cruise. We acknowledge the EMME-CARE project from the European Union's Horizon 2020 Research and Innovation Programme (grant agreement No. 856612), as well as matching co-funding by the Government of the Republic of Cyprus.

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
