# Peer review of "Shipborne measurements of methane and carbon dioxide in the Middle East and Mediterranean areas and contribution from oil and gas emissions"

_Atmospheric Chemistry and Physics, 2021_

## Author Response (AR1)

**Point-by-point response to all referee comments**
acp-2021-114

Anonymous Referee #1

**Overall Evaluation**

This manuscript utilizes an excellent dataset and derives many conclusions, particularly highlighting the discrepancies between measurements and poorly constrained inventories in these regions.

The measurements section requires some extra detail for clarity, and the derivation of the time-averaged $CO_2$ and $CH_4$ data for direct correlation with NMHC GC data needs explanation.

We thank Reviewer 1 for these very useful comments that strongly helped improving the paper and its conclusions.

We address these comments in more detail below (see Specific comments).

For the modelling some more consideration needs to be given to the predominance of the waste and agricultural sources along a narrow coastal strip of the Arabian Peninsula and closer to the ship track, while many O & G emissions, particularly oil extraction activities, are further inland.

At the scale of the campaign and within the main ship route of the Arabian Gulf where measurements were performed, the model resolution does not allow a more detailed spatial investigation of sources than the one performed here, while our complementary alkane approach does not provide directly a clear signature from other (non O&G) sources.

*No change in ms*

**Specific Comments**

2.2 Measurements Line 125 – water vapour contents were corrected – up to what % was recorded at the sub-tropical latitudes? Is the correction linear to this level?

Humidity up to 4% was recorded by the Picarro instrument at subtropical latitudes, which is indeed beyond the typical range of water vapour expected for the instrument.

Picarro analyzers calculate a correction factor Cr as a quadratic function according to Eq. 1:

$$Cr = Cwet/Cdry = 1 + I1 * H2Or + I2 * H2O^2 \text{ (Eq. 1)}$$

For this particular instrument, the quadratic factory water vapor correction had been overridden prior to the campaign to fit the ICOS Metrological Laboratory bench tests shown on the Figure below. The values are given in Table 1.

**Table 1. Revised values for correction of water vapor on CO2 and CH4**

| H$_2$O correction coefficients | CO$_2$ | CH$_4$ |
|---|---|---|
| I1 | -1.212e-02 | -1.023e-02 |
| I2 | -2.213e-04 | -9.654e-05 |

The difference between corrected and uncorrected measurements are shown in the Figure below. At 4% humidity, the extrapolated CH4 correction is -0.8 ppb, to be compared to 0.6 ppb correction at 3% humidity. For CO2, we calculated a correction of 0.20 ppm at 4% vs 0.15 ppm at 3%. The extrapolation beyond 3% humidity does not occur in a strongly non-linear range of the correction function. We are therefore confident that the impact of potential errors in the water vapor correction at high humidity has no significant effect on our results.

We added a short summary of these elements in the manuscript text:

L149: added: (water vapor effects) were determined specifically for this instrument…

2.2 Line 129 – Where do these calibration gases come from and which instrument is capable of measuring CH₄ and CO in ppb to 3 decimal places? Don't give these to more decimal places than the specified instrument precision.

These calibration gases are ambient air grease-less compressed (RIX compressor) in Luxfer tanks and calibrated against primary WMO standards on a reference Picarro instrument in the lab.

We have revised the significant digits provided in the manuscript. We now provide 2 digit decimals for CO2 reference gases and 1 digit decimal for CH4 and CO.

L152: added: Calibration gases were prepared in Luxfer aluminum cylinders

2.2 Line 132 - Precisions for G2401 are based on temperature-controlled lab conditions, not a moving platform at different temps. What was the precision for the on-ship calibration runs.

On the ship, calibration injections were quite stable and consistently exhibited a within-run standard deviation of 0.03ppm $CO_2$, 0.3ppb $CH_4$ and 10ppb CO with little change during the trip.

L167: added Calibrations were quite stable and consistently exhibited a within-run standard deviation of 0.03 ppm $CO_2$, 0.3 ppb $CH_4$ and 10 ppb CO with little change during the trip.

Line 190 – Is this each region as defined in Fig. 1?

Yes.

L236 We have added "See Fig. 1 for region definition." In fig. 3 caption for clarity.

Section 3.2 – Through to the title of this section the only mention is of alkanes, then in this paragraph it suddenly switches to NMHC. Please be consistent, or clarify that you are referring to the same dataset for these measurements.

This is indeed the same dataset. We now refer consistently to alkanes in the revised MS.

NMHC replaced by alkanes in the ms when referring to our dataset.

Fig. 5 – 4 main categories, but only 3 trend lines that don't necessarily seem to relate to the data. Which category is the lime-green trend line referring to? Needs more explanation in the caption. This figure compares ethane from GC measurements of NMHC with $CH_4$, presumably from Picarro measurements. How were the values of $CH_4$ derived to make these direct correlations?

The figure was indeed complicated to interpret given the variety of colors. The colors are now harmonized so that regression line color matches the point color by region.

Comparison between Picarro data ($CO_2$, $CH_4$) and NMHCs requires aligning time steps of the measurement systems. GC-FID system typically samples air for 10-20 minutes before injection, with a new sample being collected 30 minutes later. Prior to being compared to the GC data, $CH_4$ and $CO_2$ are averaged over exactly the 10-20 minutes of air sampling of the GC-FID, therefore also ignoring the 20 following minutes.

Fig 5 has been modified accordingly
L266: We add a sentence at the end of the 1$^{\cdot}$ paragraph in sect. 3.2. stating: "In this analysis, all $CH_4$ and $CO_2$ measurements are averaged over the sampling interval of the GC-FID."

Fig.6 – Same comment as above. How are the averaged $CO_2$ and $CH_4$ values derived for comparison with the NMHC measurements?

Same as above.

3.4 Were ship engines turned off for the whole time in the harbours as no obvious evidence for filtered data?

The engine is typically idling in harbours. Figure 8 shows CH4 data where ship stack contamination is flagged out only when the ship is moving. In harbours, all data are kept. We show all the data but refrain from quantitative analysis in harbors and focus rather on open sea measurements that represent better the larger scale.

L396: added to Fig 8 caption All data at harbours are included.

Line 339 – Kuwait City has no natural gas network, but numerous large landfill sites. Inventory emissions from O & G activity are far-removed from the coastal region. Is being so close to big sources a good comparison with modelled inventory data? These sources would potentially be within the same 01 x 0.1° model box as the receptor point.

We agree with the reviewer's concern.

L404 We added a sentence to make this point explicitly, stating: "However, due to the proximity of sources within the inventory spatial resolution the model may not resolve fully the ratio between these contributions."

3.5 What about photochemical ages for the times spent in harbours? Does this suggest an even smaller source footprint than 38 km? Harbours are often in industrial areas, potentially with storage of fossil fuels and very local generation of emissions that strongly influence the observed peaks.

We thank the reviewer for this valuable suggestion. However, the equation yields unrealistic (slightly negative) values for Kuwait harbor only. This is probably due to the poor representation of the propane to pentane ratio for Kuwait alone in the inventory: the pentane to propane ratio is lower in the Kuwait inventory (0.389 mol mol-1) compared to the Kuwait harbor measurement (regression 0.478 mol mol-1) and the model expect the ratio to decrease with time starting from the inventory value. Therefore, a harbor analysis would require a level of resolution and precision in the inventory that is not available here.

No change in ms

Line 425 – As pointed out the wind fields are at approximately 100 x 100 km scale. The winds in the north part of the Arabian Gulf are highly variable. On land the winds are dominantly from the NW. In the Gulf they often blow from the SE and these sea breezes lead to unpredictable emission dispersion along the coastal strip that may be difficult for model simulation. Less than 5 km inland the wind can be 180° different to along the coastal strip. HySplit simulations, for example, are unable to pick up these sea breezes.

We thank the reviewer for this insight.

L503 We have added the following sentence to reflect on this potential limitation: "Capturing local sea breeze patterns might be challenging for the model's driving wind field. "

**Technical Corrections**

Fig. 8 – Need to reposition geographic labels that don't fit into the boxes. Lot of detail so probably better as a full-page figure.

We agree that the figure needs to be full page.

Fig 8 We have improved the figure in terms of geographic labels.

Line 354 – should be naphtha.

L431 We corrected the spelling accordingly

Line 405 – paragraph is about $CH_4$, but refers to Fig. 9 that is about $CO_2$

Thanks. This should be referring to Fig 8
L483 it has been corrected accordingly.
* * *
**Anonymous Referee #2**

Important study presenting CH4 and CO2 data for a key oil and gas production region – where there exists limited measurement-based characterization of emissions.

The authors present a thorough characterization of CH4 and CO2 enhancements, and simulate potential sources of emissions. The paper would be benefit from a discussion on what would be needed as next steps in terms of fully quantifying emission rates (not only characterizing the mixing ratios), this is important as towards the end the authors ask if they can verify inventories.

We thank reviewer 2 for his useful insight and comments that greatly helped enhancing the paper.

We modified the end of our conclusion to add more elements about future needs.

L556 and following We now state that: "More targeted measurements investigating specifically offshore and onshore extraction sites will enable a better understanding of the

distribution of emission in the area. Assessing emissions from individual wells and processing facilities with dedicated measurements and combining these estimates at the regional level would be necessary to improve further our knowledge of actual O&G emissions in the Middle East. Combining $CH_4$ and alkane measurements offers the potential to 1) separate the O&G component of measured $CH_4$ from other sources and 2) to investigate at the regional and local level the fraction of emissions linked to each phase of O&G value chain, from extraction to end-usage through storage, transport and processing. "

Line 3: change the term 'atmospheric distribution' or expand descrption of this term.

We simplified the sentence by referring to atmospheric mixing ratio.

L14: "We measured the atmospheric mixing ratios of $CH_4$ and $CO_2$ by ship''

Introduction

Line 25: Highlight short-term potency of methane. Importance to illustrate the difference in their climate impact.

We now mention the 28x higher GWP over 100 year. We also added mention of the efficient and rapid mitigation options available for CH4, referring to the recent study by Ocko et al., 2021 in the following paragraph.

L25: added: Over a 100-year horizon, $CH_4$ has a global warming potential 28 times larger than $CO_2$.

L48: added: "Rapid and efficient mitigation options are readily available for methane emissions. 80% of economically feasible abatement reside in the oil and gas (O&G) sector (Ocko et al., 2021).

Lines 40-43: Important to highlight that the fossil fuel sector haskey mitigation opportunities, likely more cost-effective than the other sectors (e.g. waste or ag). You could cite IEA methane tracker work and opportunity to reduce a significant fraction of emissions at net zero cost. I would suggest the IEA data instead of the MARCOGAZ report.

We now highlight more specifically the importance of oil and gas mitigation options and their cost-effectiveness. In addition to Ocko et al., (2021) we also added reference to IEA (2020), Methane Tracker 2020, IEA, Paris https://www.iea.org/reports/methane-tracker-2020

L48: added: "Rapid and efficient mitigation options are readily available for methane emissions. 80% of economically feasible abatement reside in the oil and gas (O&G) sector (Ocko et al., 2021). IEA (2020) reviewed a number of efficient, cost-effective abatement options for O&G sector and found that 40% of O&G methane emissions could be avoided at a zero net cost."

Line 46. Alvarez et al is relevant here but this is only for the US. Similarly, I would suggest rephrasing that methane emissions occur throughout the oil and gas supply chain. Alvarez et al. also shows that in the US majority of emissions are for upstream sector.

We added this clarification that Alvarez et al. (2016) was specifically for the US, and added a sentence on where along the value chain did CH4 emissions occur for the oil and gas industry respectively.

L54 we changed to : "For the oil industry, methane emissions occur essentially as indirect emissions, as venting or incomplete combustion during flaring and during transport and refining (IEA, 2020). For the gas industry, emission of $CH_4$ occurs at all stage of the life cycle as fugitive emission (leaks from valves, connectors and compressors, intentional venting) but also as incomplete combustion during flaring (GIE-MARCOGAZ, 2019)."

L57 We specified that: "Alvarez et al. (2018) found that in the United States a large fraction of the net emission is associated with the production, transport and processing."

Line 55. Any reason for not mentioning Hmiel et al. here?

We add a sentence stating that: "based on ice core $^{13}CH_4$ measurements, Hmiel et al. (2020) found that fossil fuel methane sources could be underestimated by as much as 25-40%."

L64: added : "Based on ice core $^{13}CH_4$ measurements, Hmiel et al. (2020) found that fossil fuel methane sources could be underestimated by as much as 25-40%."

Suggest revising consistent use of significant figures throughout the manuscript.

We reviewed the consistency of decimal use throughout the manuscript and reduced the number of digits for emission data especially in the introduction.

Done throughout the ms

Line 69. For fields discovered in Levantine Sea, are they already producing? If so, is production significant?

Gas exploitation occurs essentially in Egyptian waters. Egypt represents 1.5% of global natural gas production in 2018 (BP statistical review of world energy 2019). The Leviathan gas field off the coast of Israel is reported to have started production 31 December 2019. Most of the activity in other countries of the region remains as exploration currently in the area, with geopolitical interferences limiting the ability to commercially exploit some of the fields.

No change in ms

Line 70. I assume that these emissions estimates are from EDGAR, please mention explicitly. Can you also include estimate from UNFCCC. For this, you can use Scarpelli et al.

We now indicate explicitly when the EDGAR database was used. To reflect on the reviewers' hint at difference between inventories, we report based on Global Carbon Project numbers that: "Across the different inventories reported in Saunois et al. (2020) for the Middle East, the spread represents 18% of the mean emission intensity, reflecting a significant uncertainty on country-level emissions". Scarpelli et al. do not provide explicitly country-based emissions.

L90: added explicitly "(reported) by the EDGAR inventory"

L92: added "Across the different inventories reported in Saunois et al. (2020) for the Middle East, the spread represents 18% of the mean emission intensity, reflecting a significant uncertainty on country-level emissions."

Line 95. In terms of referencing Yacovitch et al. Can you expand on why you do not attempt to estimate emission rates?

This is a very good point. Our cruise route was planned ahead along the main ship corridor and did not target specifically oil and gas exploitation. Therefore, unlike Yacovitch et al., we lacked the ability to investigate extensively and specifically well-identified oil and gas wells/platforms, and assess emission factors on an individual site basis. We reflect on this point in the outlook discussion and add to the conclusion that "Assessing emissions from individual wells with dedicated measurements and combining these estimates at the regional level would be necessary to improve further our knowledge of actual O&G emissions.".

L559: added "Assessing emissions from individual wells and processing facilities with dedicated measurements and combining these estimates at the regional level would be necessary to improve further our knowledge of actual O&G emissions in the Middle East."

Lines 160-170 I suggest that you also compare your inventory results to the Scarpelli et al. gridded inventory, as this is based on UNFCCC data.

We thank the reviewer for this valuable suggestion. Indeed, the choices made in the EDGAR inventory (global consistency of emission factors and activity data) is markedly different from the Scarpelli et al. approach that ensure country totals to be consistent with UNFCC reporting. Here we use the two different inventories to assess in a crude way the sensitivity of our simulations to change of inventory. A full comparison of all available inventory data in model-data comparison is outside the scope of this campaign-based paper and will form the basis of another study.

No change in this ms

Also, do you adjust EDGAR to any changes in production during the time of the study?

We used directly the 2012 emission data without adjustment for time variation. The impact on this choice on our result is already discussed on lines 438 and following.

No change in ms

Figure 4- Include units of radiative heat.

We add this unit (MW) in the figure caption.

L252: added (in MW)

Line 220 – I would caution (or request expanding) discussion on using flaring as proxy for extraction and production sites. While this can be true, gas production fields tend to have less flaring. At the same time, oil production fields could be venting gas instead of flaring.

In the section starting line 220 we do not use flaring as a proxy. Fig. 4 immediately above presents flaring points only a "proxy for the presence of extraction and production site" to illustrate the proximity of the ship track at regional scale with potential emitting areas.

No change in ms

Line 239 – Emissions could also be related to venting, not necessarily fugitive emissions. Also, what about the correlation between CH4 and CO2 to check for combustion sources (not only correlation with NMHCs).

We thank the reviewer for this useful suggestion. In the Table 2 below, correlation between CH4 and CO2, as well as CH4 and CO correlations, are given for each region discussed in Section 3.2

| Region | Correlation with $CO_2$ | P value | Correlation with CO | P value |
|--------|------------------------|---------|--------------------|---------|
| Arabian Gulf | 0.53 | 1.976e-12 | 0.76 | < 2.2e-16 |
| Oman Gulf | -0.22 | 0.03267 | 0.88 | < 2.2e-16 |
| Red Sea | -0.08 | 0.1368 | 0.23 | 2.011e-05 |
| Suez Canal | 0.83 | < 2.2e-16 | 0.87 | < 2.2e-16 |
| | | | | |

Considering these numbers we revise our sentence on potential sources in the Arabian Gulf and improved this section.

L276-289 correlations are added and some changes are implemented in line with the reviewer's point. The modified text now reads: "$CH_4$ was also correlated with $CO_2$ ($r = 0.53$, $p = 2 \cdot 10^{-12}$) and with CO ($r = 0.76$, $p < 10^{-15}$) in the Arabian gulf, suggesting that $CH_4$ variability is dominated by emissions from O&G activities collocated with combustion sources. Flaring may typically contribute to such an emission profile. However, $CO_2$ showed no significant correlation with any of the three alkanes mentioned above. This suggests that $CO_2$ variability in the area is dominated by other factors than O&G sources.
Over the Red Sea, $CH_4$ enhancements of about 200 ppb above background have been observed. A clear correlation between $C_2H_6$ and $CH_4$ ($r = 0.71$, $p < 10^{-15}$) is associated with a regression slope of 0.047 ppb ppb$^{-1}$. No significant correlation is found for $CH_4$ against $CO_2$ ($r = -0.08$, $p = 0.1368$) or CO ($r = 0.23$, $p = 2 \cdot 10^{-15}$).
For the Suez Canal area, the regression slope between $C_2H_6$ and $CH_4$ is 0.006 ppb ppb$^{-1}$ ($r = 0.75$, $p < 10^{-15}$). Strong correlations are found between $CH_4$ and $CO_2$ ($r = 0.83$, $p < 10^{-15}$) and

CH$_4$ and CO ($r$ = 0.87, $p$ < 10$^{-15}$). This indicates that combustion sources are strongly contributing to CH$_4$ variability in the Suez Canal. The airmass originates from the Eastern Mediterranean and Egypt according to Flexpart backtrajectory simulations. Differences in C$_2$H$_6$ to CH$_4$ regression slopes between the Suez Canal and the Red Sea may therefore reflect either an aggregation of different O&G emission sources or different fraction of O&G in observed CH$_4$ enhancements. "

Line 255: methane to ethane ratio depends on gas composition, but also on source of emissions. Ratio would be different if emissions are happening at the wellhead, at a storage tank, or after a processing plant.

We thank the reviewer for this remark to which we agree. We add accordingly a notion that the ratio changes "along the production chain"

L307 added "along the production chain"

Line 260: For natural gas the ratio is expected to be 0.86 (indicated as horizontal line in Fig. 6). Can you expand on why is this expected for natural gas? Is this for natural gas production stage? What about other stages of the supply chain? And different gas compositions?

According to Gilman et al. (2013) the i- to n-Pentane isomeric ratio appears to be robust across different oil and gas fields (they mention 0.89 in Russia's Kola peninsula, 0.86 in Wattenberg area, Colorado; 0.85 for natural gas in the North Sea, 0.82 for Macondo reservoir). The mean value is determined experimentally (see references in Gilman et al., 2013; and Bourtsoukidis et al., 2019). The ratio appears to be stable across oil and gas fields. This ratio also has the advantage of limited sensitivity to photochemical oxidation in the atmosphere.

Isomerization of oil after distillation is widely used to obtain gasoline with higher octane indices. This process results in a decreased nC5 concentration in gasoline and hence in a higher iC5/nC5 ratio in urban environment. In natural gas processing, pentane is part of the condensate separated from the commercial natural gas.

Therefore, iC5/nC5 ratios found close to the empirical range of 0.8-0.9 associated to high methane concentrations can point to methane emission that occurred at any point between the well and the refinery/natural gas processing plant.

We add explanations in the manuscript and clarify the paragraph's concluding sentence ("clear dominance of sources linked to O&G extraction and production.")

L320 we added: "Isomerization of oil after distillation is widely used to obtain gasoline with higher octane indices. This process results in a decreased nC5 concentration in gasoline and hence in a higher iC5/nC5 ratio in urban environment. In natural gas processing, pentane is part of the condensate separated from the commercial natural gas."

L324: the revised text now reads: "…clear dominance of sources linked to O&G extraction and production"

Line 320; Can you expand on known source locations? Is this only based on EDGAR? I would expect high density of oil and gas infrastructure relative to granularity of EDGAR inventory. Also, can you expand on how episodic emissions could be impacting your simulations (i.e. impact of super-emitters)?

We now correct the formulation. We were actually referring to the position of the atmospheric enhancements as observed along the ship track. We modify the text accordingly.

We agree that the resolution in EDGAR does not allow to identify individual source position, and, given the smoothing effect of the gridding of inventory, even less to simulate the impact of nearby (<0.1°) individual super-emitters.

L381: the sentence now reads "poorly positioned along the ship track." (we removed the irrelevant reference to "known source location")

Lines 340-345: It would be useful to expand on potential differences in methane emissions between oil vs gas production (or combined production). It could be hypothesized that gas fields (where natural gas in main product) could have lower emissions that oil fields (where associated gas is a co-product, often not captured).

We fully agree with this comment. In a wide US survey, Lyon et al. 2016 found that superemitter sites (>1-3g/s) were 3 times more frequently associated to oil than the gas assets, and 90% of sources were from "tank vents and hatches". Our study would support an underestimation of these emissions by the inventories, and an overestimation of leak rates in the upstream natural gas industry. We added these sentences to the revised text.

L415 we added: "In a wide US survey of 8000 facilities, Lyon et al. (2016) found that super-emitter sites were 3 times more frequently associated to oil than the gas assets, and 90% of sources were from tank vents and hatches used in oil storage. Our study would suggest an underestimation of oil upstream emissions by the inventories, and an overestimation of leak rates in the upstream natural gas industry."

Ref: Lyon, D. R., Alvarez, R. A., Zavala-Araiza, D., Brandt, A. R., Jack- son, R. B., and Hamburg, S. P.: Aerial Surveys of Elevated Hydrocarbon Emissions from Oil and Gas Production Sites, Environ. Sci. Technol., 50, 4877–4886, doi:10.1021/acs.est.6b00705, 2016.

Can you also expand on influence of onshore infrastructure (processing gas from the offshore platforms) vs emissions from offshore infrastructure?

Oil and gas well occur onshore as well as offshore, while processing plants remain only onshore. The pentane isomeric ratio has the ability to separate upstream emissions of methane (as associated or non-associated gas) from emissions from refined/processed products. However, the model is limited by the challenge of poorly represented sea breeze effects and by the resolution of the inventory in this highly imbricated area.

L503 we added : Capturing local sea breeze patterns might be challenging for the model's driving wind field.

L556 in the conclusion we added: "Separating on-shore from off-shore emissions would require a dedicated modelling study investigating local atmospheric circulation combined with atmospheric composition measurements."

---

## Referee Report (RR1)

**Paris et al. - Shipborne measurements of methane and carbon dioxide in the Middle East and Mediterranean areas and contribution from oil and gas emissions – Post-Review**

**Overall Evaluation**

The authors have answered my questions and comments raised in pre- and full review and modified the manuscript accordingly. They have taken onboard the small edits / corrections that were suggested to provide clarity to specific aspects of the manuscript. I do not believe that the manuscript requires further modification.

---

## Author Response (AR2)

1 -- Show Flexpart trajectories analysis figures to help comprehension of the discussion

➔ This is now added in Appendix Fig. A3, with 3 representative footprints selected to support the discussion. This figure is cited where appropriate in the main text. (L193, 214, 262)

2- L163 - spell ECMWF

➔ Done at the first occurrence L 163

3- Fig.2 - Add more information about Leg1 and Leg 2 in the subtitle

➔ Done in caption of Fig. 2 L199.

4- show maps of the model EDGAR

➔ 2  maps are added in  Appendix, Fig A1 and A2 for $CH_4$ and $CO_2$ respectively. They are referenced in the text L 174.

Please note that a few very minor editions for consistency, style and typo have been made :
- L158 (add: unless specified otherwise),
- L191 (remove hyphen),
- L364 (replace retroplume by backtrajectory),
- L426 (correct date -> data)